# Active Microbiological Surveillance for Contrasting Multi-Drug-Resistant Pathogens: Comparison Between a Multiplex Real-Time PCR Method and Culture

**DOI:** 10.3390/diagnostics15172128

**Published:** 2025-08-22

**Authors:** Gaetano Maugeri, Maddalena Calvo, Guido Scalia, Stefania Stefani

**Affiliations:** 1Department of Biomedical and Biotechnological Sciences (BIOMETEC), University of Catania, 95123 Catania, Italy; gmaugeri88@gmail.com (G.M.); lido@unict.it (G.S.); stefania.stefani@unict.it (S.S.); 2U.O.C. Laboratory Analysis Unit, A.O.U. “Policlinico-San Marco”, 95123 Catania, Italy

**Keywords:** multi-drug-resistant pathogens, microbiological surveillance, multiplex real-time PCR, carbapenemases, CTX-M, *van* genes

## Abstract

**Background/Objectives**. Multi-drug-resistant (MDR) microorganisms pose a significant challenge in healthcare settings, particularly with beta-lactam-resistant Gram-negative bacteria and glycopeptide-resistant enterococci. Culture represents the most reliable technique in determining their presence within surveillance swabs. However, it requires a long time-to-result (TTR) and shows low sensitivity. Molecular techniques integrate diagnostic procedures, allowing TTR reduction and precise identification of genes. **Methods**. During our usual surveillance campaign, we had the opportunity to evaluate the Allplex Entero-DR assay (Seegene Inc., Seoul, Republic of Korea) and the Entero-DR Plus assay (Arrow Diagnostics srl, Genova, Italy) molecular kits for the detection of extended-β-lactamases (ESBL), carbapenem- and vancomycin-resistant genes, as well as *Acinetobacter* spp. and *Pseudomonas aeruginosa* spp. identification directly from rectal swabs. A comparison between these tests and the culture-based routine completed the study. **Results**. The analysis included 300 rectal swabs from the University Hospital Policlinico (Catania, Italy). One hundred and eighty-eight samples (62.6%) resulted as positive for at least one Allplex™ target, reaching optimal sensitivity and negative predictive value (100%). Our results underlined the ubiquitous *bla*CTX-M and *van* genes presence and demonstrated the diffusion of double-carbapenemases genes and metallo-β-lactamases-producing strains. In our epidemiological setting, few data were collected about carbapenem-resistant *P. aeruginosa* and *Acinetobacter* spp., which require further evaluations on simultaneous respiratory colonization and higher sample numbers. **Conclusions**. Our analysis highlighted the importance of combining conventional and advanced diagnostic methods in investigating MDR pathogens. The right approach should be based on the prevalence and variability of resistance mechanisms within a specific epidemiological area. Remarkably, molecular screenings may exclude negative samples within high-risk areas due to a significant negative predictive value.

## 1. Introduction

The spread of multi-drug-resistant (MDR) microorganisms is one of the most challenging issues among critical healthcare settings, particularly concerning carbapenem-resistant Gram-negatives and glycopeptides-resistant enterococci [1]. International travel, migration, and tourism contributed to MDR cross-diffusion within European countries, intensifying epidemiological surveys [2]. The latest European Center for Disease and Control Prevention (ECDC) report registered concerning resistance percentages for these pathogens, especially regarding Italy and the National Institute of Health data [3]. Particularly, Sicily accounted for 88.2% of carbapenem-resistant *Acinetobacter* species and 40.7% of carbapenem-resistant *Klebsiella pneumoniae*, which were frequently isolated from severe infection episodes [4]. Additionally, carbapenem-resistant *Pseudomonas aeruginosa* (14.6%) and carbapenem-resistant *Escherichia coli* (2%) have emerged as systemic or nervous central system infections aetiological agents within the same geographical area.

Among Gram-positive microorganisms, the glycopeptide resistance rate deserved increasing attention, registering 11.9% of vancomycin-resistant *Enterococcus* species in Sicily from systemic infections [4]. According to these results, surveillance programmes are fundamental across high-risk epidemiological areas, reporting rectal swabs as an ideal specimen due to simple collection mode and timing. Although there is a homogeneous consensus on the importance of rectal swabs for multi-drug-resistant pathogens alertness [5,6,7], a widespread discussion exists in the applied methodology. Guidelines and the literature consider the culture-based method the most reproducible and reliable technique in detecting MDR microorganisms, allowing identification of colonies and antimicrobial susceptibility testing with minimum inhibition concentration (MIC) values [8,9]. Despite these fundamental characteristics, cultures require a long turn-around time (TAT) and lose sensitivity depending on the tested biological sample [5]. These assays do not investigate specific resistance mechanisms and do not detect low-level expression of carbapenemases showing susceptibility MIC values [1]. Moreover, cultures continue to be considered the gold standard in comparison with new methodologies [8,9].

However, scientific reports have highlighted the importance of distinguishing resistance genes due to the increasing presence of enzyme variants and double-resistance markers within the same pathogen [10]. Molecular techniques have integrated microbiological diagnostic procedures for decades, allowing TAT reduction and precise single or multiple resistance genes identification [1,2,3,11,12]. The current commercial spectrum proposes numerous molecular kits in providing resistance markers detection directly from biological samples, collecting encouraging comparisons to standard culture methods and faster patient management within critical hospital settings [1,2,3,10,11,12]. Herein, we propose an experimental evaluation of the Allplex™ Entero-DR assay (Seegene Inc., Seoul, Republic of Korea) and Entero-DR Plus assay (Arrow Diagnostics srl, Genova, Italy) multiplex Real-Time quantitative polymerase chain reactions (qPCR) in detecting β-lactams- and vancomycin-resistance genes directly from rectal swabs. The final purpose was to emphasize the importance of a rapid support protocol in antimicrobial resistance monitoring, contextualizing its results within specific high-risk units and comparing them to conventional detection methods.

## 2. Materials and Methods

### 2.1. Study Design and General Information

The experimental analysis was completed at the University Hospital Policlinico (Catania, Italy) during a six-month evaluation (January–June 2024), including routinary rectal swabs surveillance from Haematology, Intensive Care, Internal Medicine, Emergency Room, Pneumology, Surgery, Transplants, and Paediatric units patients. All of these hospital wards included a rectal screening for MDR microorganisms on admission of patients, weekly during their recovery, and before discharge. This active surveillance strategy depended on the previously reported high MDR prevalence within the above-mentioned units. Clinicians collected rectal samples using the Copan Faecal Swab^TM^, containing the liquid Cary–Blair medium for biological specimens collection, storage, and transportation. Our protocol involved these samples avoiding any supplementary biological material request. Specifically, the rectal swabs simultaneously experienced conventional diagnostic procedures (culture-based method) and experimental molecular workflow through the Allplex™ Entero-DR assay (Seegene Inc., Seoul, Republic of Korea) and Entero-DR assay Plus (Arrow Diagnostics srl, Genova, Italy). The study was conducted according to the guidelines of the Declaration of Helsinki and the best clinical practice (D.M. 15 July 1997). The present study does not directly involve patient management or drug administration. The studies were conducted in accordance with the local legislation and institutional requirements. Figure 1 summarizes the applied experimental protocol.

### 2.2. Culture-Based Method

We inoculated 10 μL of liquid rectal swabs transport medium into a MacConkey agar plate (Vacutest Kima srl, Arzergrande, Italy), adding 10 μg meropenem and 30 μg ceftazidime disks (Liofilchem**^®^**, Roseto degli Abruzzi, Italy) as an indicator for possible Gram-negatives extended-spectrum-β-lactamases and carbapenemases production. Furthermore, a second aliquot of 10 μL was inoculated into tryptone soy agar supplemented with 5% of sheep blood (Liofilchem**^®^**, Roseto degli Abruzzi, Italy), including a 30 μg vancomycin disk for vancomycin-resistant enterococci detection. The agar plates underwent a 37 °C overnight incubation period, and the eventually grown colonies proceeded to identification and antimicrobial susceptibility testing only in the case of β-lactam and/or vancomycin resistance suspicion. Overall, the above-mentioned culture protocol derived from previously published studies, including some modifications such as the vancomycin discs’ addition [8,13,14]. The eventual meropenem or vancomycin inhibition diameters were evaluated according to the European Committee on Antimicrobial Susceptibility Testing (EUCAST) guidelines [15].

The MALDI-TOF MS Biotyper Sirius system (Bruker Daltonics, Billerica, MA, USA) identified the β-lactam- and/or vancomycin-resistant strains, while the VITEK**^®^** AST-439 and AST-658 cards (Biomerieux, Florence, Italy) provided antimicrobial susceptibility testing according to the EUCAST document [16]. The Kirby–Bauer method determined the cefiderocol MIC values for some included Gram-negative strains.

Finally, the Gradient test method integrated the aztreonam/avibactam susceptibility testing for metallo-β-lactamases-producing strains. The phenotypically beta-lactams-resistant strains grown colonies faced a supplementary molecular process to detect KPC, OXA48, NDM, VIM, or IMP genes, requiring 50–60 min by Cepheid Xpert**^®^** Carba-R cartridges. Otherwise, the phenotypically glycopeptides-resistant strains grown colonies underwent a supplementary immunochromatographic method to detect the CTX-M enzyme, lasting 15 min by RESIST CTX-M (Coris BioConcept, Gembloux, Belgium). The global conventional protocol required at least 48 h to furnish a definitive result.

### 2.3. Real-Time PCR Analysis

The Allplex™ Entero-DR assay (Seegene Inc., Seoul, Republic of Korea) and Entero-DR Plus assay (Arrow Diagnostics srl, Genova, Italy) are qualitative multiplex real-time PCR kits. Specifically, the Entero-DR assay kit can reveal *bla*_KPC_, *bla*_OXA-48_, *bla*_VIM_, *bla*_NDM_, *bla*_IMP_, *bla*_CTX-M_, *vanA*, and *vanB* genes. According to the commercial validation, carbapenemases genes may be related to *Enterobacterales* or *Pseudomonas* spp. On the other hand, the Entero-DR Plus assay detects *Acinetobacter* spp. and *Pseudomonas* spp. These identifications may be matched to carbapenem-resistance markers detected through the Entero-DR assay mix, which is the only one to have been released for official diagnostic usage. The molecular assay started with an automated nucleic acid extraction and PCR setup through the Seegene STARlet, requiring 2 h and 45 min. The Allplex™ Entero-DR assay and Entero-DR Plus assay amplifications were carried out on the Bio-Rad CFX96 thermal cycler module, lasting 1 h and 15 min. Anonymized results were collected into a worksheet, including sample collection day, ward, rectal swab phenotypic result, and β-lactams and glycopeptides MIC values. Allplex results were categorized depending on the involved patient. In particular, the following areas were focused on:The identification of carbapenem-resistant *Enterobacterales*, *van* genes, and *bla*_CTX-M_ genes was reported for all the included patients, considering the high dissemination risk from gastrointestinal colonization among critical patients.The identification of *Acinetobacter* spp. and *Pseudomonas* spp. without any resistance markers from the first reagent kit was reported only for intensive care patients, especially in the case of at least three colonized anatomical sites, due to the risk of related respiratory (i.e., ventilator-associated pneumonia) or systemic infections.The identification of *Acinetobacter* spp. and *Pseudomonas* spp. associated with resistance markers from the first reagent kit was reported for all the involved patients to limit the resistant strains’ spread among all the wards.

### 2.4. Data Comparison and Statistical Analysis

Discrepancies were defined as data different from a phenotypic result (for instance, a molecular negative result in the case of a phenotypic resistance MIC value or a molecular positive result in the case of a phenotypic susceptible MIC value), which is the gold standard method for alert microorganisms detection. The culture-based methods furnished the identified species whose eventual resistance markers were previously detected by the molecular method. The eventual discrepancies between the Allplex procedure and conventional results were further investigated through the Xpert Carba-R assay (Cepheid, Sunnyvale, CA, USA) which is already integrated into our laboratory routine as a molecular confirmation system. Sensitivity, specificity, positive predictive value, and negative predictive value were calculated to complete the comparative analysis. Finally, we calculated a Chi-square test result (significance level at 0.05) to investigate eventual discrepancies between the observed and the expected results.

## 3. Results

Our experimental evaluation involved 300 rectal swabs from different hospital wards. Specifically, 188 samples (62.7%) resulted positive for at least one Allplex target, while 112 swabs (37.3%) tested negative for all the included molecular targets. Positive and negative results emerged differently among the involved hospital units, depending on the patients’ complications and number of samples. Figure 2 summarizes this distribution of results. On the other hand, culture-based methods reported 68 positive results (22.7%) and 232 (77.3%) negative samples.

As regards carbapenemases-producing strains, the Allplex^TM^ Entero-DR molecular assay detected 55 samples (18.3%) positive for at least one carbapenemases gene. A total of 29 samples (52.8%) reported positive cultures for carbapenemases-producing bacteria, which were further confirmed by the Cepheid Xpert Carba-R system. Otherwise, 26 samples (47.2%) revealed a negative culture result. Molecular and culture-based results were compared, and Table 1 illustrates statistical parameters for carbapenemases genes detection.

The Allplex molecular analysis reported 118 (39.3%) CTX-M positive results, accounting for 111 (94.1%) culture-negative results. On the other hand, seven samples (5.9%) reported a culture-positive result for ESBL-producing Gram-negative strains. Specifically, two samples (28.6%) showed ESBL-producing *K. pneumoniae*, along with ESBL-producing *Proteus mirabilis* in one sample (14.3%). Finally, one sample (14.3%) reported ESBL-producing *E. coli*. Table 2 summarizes the statistical analysis for the molecular and culture-based comparison in detecting CTX-M-producing strains.

As regards the vancomycin-resistance colonization, our analysis reported 93 (31%) *van* genes-positive results from the Allplex utilization. A number of 21 cultures (22.6%) tested positive for glycopeptides-resistant enterococci, while 72 cultures (77.4%) reported a negative result. Table 3 shows statistical details about molecular and culture assay comparison for *van* genes-carrying strains.

The Chi-square test results confirmed a consistent significance (*p* < 0.00001) among the overall Allplex Entero-DR assay results, highlighting a strong correlation between the gathered result and the applied methodology.

The Allplex Entero-DR assay frequently reported a simultaneous presence of carbapenemases marker and *van* or CTX-M genes. Table 4 summarizes all the combinations detected though this molecular method during the study period.

The Allplex Entero-DR Plus assay experimentally allowed *Acinetobacter* spp. and *Pseudomonas* spp. detection directly from rectal swabs. A total of 62 samples (20.6%) revealed *Pseudomonas* spp. molecular positive results, among which only six cultures (9.7%) showed this growth of microorganisms. Remarkably, five positive cultures (96.7%) documented carbapenem-susceptible *P. aeruginosa*, matching with Allplex-DR assay carbapenemases genes negative results. Consequently, only one isolated strain (16.6%) reported carbapenem-resistance, confirming the Allplex Entero-DR assay VIM detection on the corresponding sample. Regardless of the carbapenem susceptibility profile, 56 cultures (90.3%) tested negative for *Pseudomonas* spp. This evidence accounted for a 100% sensitivity and negative predictive value, along with a specificity of 80.9%. The positive predictive value only reached 9.7%.

As regards *Acinetobacter* spp., the Allplex Entero-DR Plus assay reported 35 (11.6%) positive results. Culture-based methods confirmed this microorganism’s presence for five cases (14.3%), accounting for 30 (85.7%) negative results. The statistical analysis documented a sensitivity of 100%, a specificity of 89.8%, a positive predictive value of 14.3%, and a negative predictive value of 100%. The Chi-square test reported a significance (*p* < 0.00001), confirming a correlation between the applied method and the obtained results. According to the gathered results, we planned different data reporting plans depending on the involved patients and the detected microorganisms. Figure 3 illustrates our data reporting flowchart after the application of both molecular and culture-based methods.

## 4. Discussion

Active surveillance strategies are strongly recommended in high MDR pathogens incidence areas [15,16]. Different microbiological surveillance strategies have been proposed to prevent the spread of these bacteria in a healthcare environment, aiming to early recognize their colonization within vulnerable human hosts [16]. Despite the elevated specificity and reproducibility of phenotypic methods, they suffer from prolonged turn-around time and low sensitivity rates. Additionally, only molecular assays identify the presence of precise genes. Microbiological surveillance programmes may include rapid molecular methods to facilitate critical patients’ screening, allowing a significant reduction in MDR prevalence. Our previous experience demonstrated how routine weekly surveillance within vulnerable patients leads to a decrease in gastro-intestinal colonization by carbapenem-resistant microorganisms [12]. According to those data, carbapenem-resistant *Enterobacterales* prevalence decreased from 37.9% (during 2021) to 31.3% (during 2022) and 5.05% (during 2023) [4].

Our study combined culture-based methods and the Allplex^TM^ Entero-DR/Entero-DR Plus assays multiplex Real-Time PCR kit in monitoring antimicrobial resistance markers within critical hospital units. The comparison demonstrated some interesting results allowing active immediate actions.

As already described in the diagnostic tool’s characteristics, all the important determinants related to carbapenem-resistance, ESBL-producing Gram-negatives, and vancomycin-resistant enterococci were included. Our results were consistent with other results reported in the literature [17]. The Allplex Entero-DR assay demonstrated optimal sensitivity and negative predictive value (100%) in detecting carbapenemases, CTX-M, and *van* genes directly from rectal swabs. As regards carbapenemases genes, we also reported a discrete specificity value. Despite these encouraging percentages, positive predictive values suffered from a significant impact due to discrepancies with culture-based methods.

Specifically, the culture was often negative for the presence of the microorganism, while the molecular methods were able to detect all the resistance genes. Previously published data registered similar disadvantages in comparing such different methodologies, especially about carbapenemases genes [11,12,17]. On the other hand, several double-carbapenemase gene combinations (45.4%) emerged from the tested samples. Most of these specific molecular results (68%) were confirmed by a positive culture and an antimicrobial susceptibility testing suggestive of carbapenem-resistant microorganisms. Carbapenemases gene amplification confirmed the previously published local literature, registering numerous metallo-β-lactamases-producing strains [10,11].

Our results demonstrated a consistent *blaCTX-M* and *van* genes diffusion in our geographical area. Remarkably, the study reported that *vanA* genes are more prevalent than *vanB* genes, and their positive cultures were confirmed by the antimicrobial susceptibility testing results. The literature lacks conspicuous data on CTX-M and *van* gene detection compared to conventional methods. Both determinants are present in species such as *Kluvyera* spp., *Pantoea* spp. (CTX-M), or *Escherichia fergusonii*, *Shigella boydii*, *Sphingobium limneticum*, and *Pseudomonas stutzeri* (for *van* genes) of the human microbiota with no clinical significance among human hosts [18,19,20]. We hypothesized that the tested molecular method could have amplified gene fragments not linked to precise species identification. This hypothesis may explain the negative culture-based surveillance result. However, discrepancies between molecular and culture methods enhanced the importance of combining resistance gene search with microorganism identification. Conversely, the molecular system can be combined with a culture method.

The early molecular application occasionally overestimated the presence of resistance markers, leading to a more conservative approach to critical hospital units. However, this approach seemed interesting regarding some specific markers. Certainly, rapid carbapenem-resistance genes detection is essential within our high-risk epidemiological area.

The elevated negative predictive value could allow an early and precise screening for negative results, optimizing the turn-around time (TAT) and patients’ cohorting within critical units. In addition to the MDR molecular screening, we tested the Allplex Entero-DR Plus assay for detecting *Acinetobacter* spp. and *Pseudomonas* spp. from rectal swabs. First of all, our hospital is characterized by a limited prevalence of both organisms. In our experience, we were not able to confirm the clinical significance of *Pseudomonas* spp. and *Acinetobacter* spp. detection within critical patients in the absence of corresponding resistance markers or proven bloodstream dissemination episodes after gastrointestinal colonization. Both microorganisms can be reported as rectal colonizers only in severe patients, especially in the absence of any carbapenem-resistance genes.

We can propose a different consideration for metallo-enzymes-producing *P. aeruginosa* and carbapenem-resistant *A. baumannii*, whose gastrointestinal colonizations should be reported for all patients. In our opinion, the molecular search for MDR *P. aeruginosa* and *A. baumannii* could be performed in other biological samples (oropharyngeal and cutaneous swabs) more related to their colonization or infection episodes. Scientific guidelines currently lack definitive specifications about the most reliable sample for active surveillance protocols.

## 5. Conclusions

In conclusion, our analysis demonstrated that active surveillance through rapid molecular screening is essential to increase local epidemiological knowledge. Molecular methods may confirm precise resistance markers or multiple genes combinations, confirming the eventual phenotypic susceptibility profile. The integration of similar technologies within our diagnostic protocols supported the tracking of colonized patients and the limitations of persistent MDR pathogens’ spread [11,21]. Moreover, the study demonstrated the importance of correctly planning microbiological surveillance protocols within healthcare settings. An active surveillance procedure may lead to an increased awareness about the healthcare-related infections rate among vulnerable and long-term hospitalized patients. Recent literature data emphasized that acquired antimicrobial resistance markers should be analysed along with virulence factors in high-risk settings, especially regarding Gram-negative bacteria [21]. Previously published papers about our epidemiological area documented hypervirulence suspicion among multi-drug-resistant Gram-negative strains [22]. On these premises, future studies may include higher number of clinical isolates, aiming to investigate both resistance mechanisms and virulence patterns through advanced diagnostic tools that are easily reliable for routine laboratory procedures.

## Figures and Tables

**Figure 1 diagnostics-15-02128-f001:**
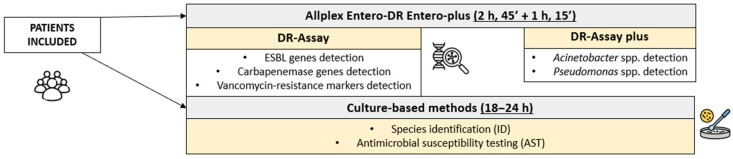
Summary of the applied protocol from patients’ inclusion to molecular and conventional method TAT comparison.

**Figure 2 diagnostics-15-02128-f002:**
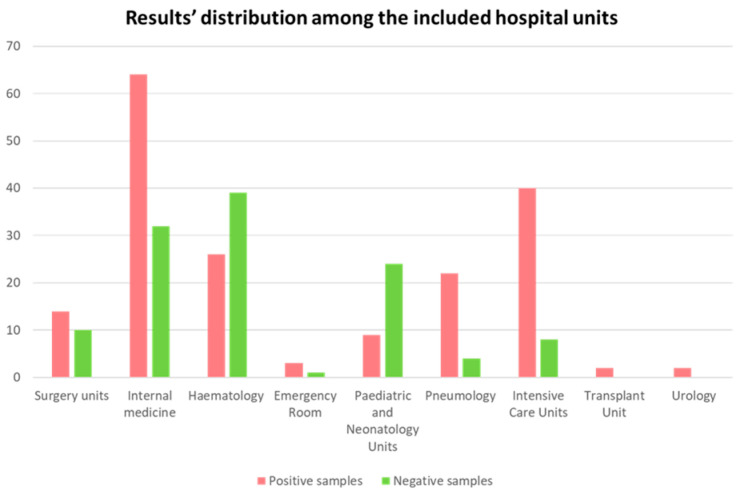
Distribution of results among the involved hospital units. This summary included all the samples positive for at least one target within the “positive samples” group. Negative samples were classified according to the absence of any molecular target.

**Figure 3 diagnostics-15-02128-f003:**
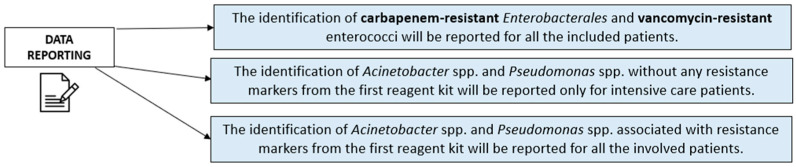
Data reporting flowchart depending on the detected microorganism and the involved patients.

**Table 1 diagnostics-15-02128-t001:** Statistical analysis for carbapenemases detection among the included samples.

Statistical Parameter	Percentage
Sensitivity	100%
Specificity	90.4%
Positive predictive value	52.7%
Negative predictive value	100%

**Table 2 diagnostics-15-02128-t002:** Statistical analysis for CTX-M detection among the included samples.

Statistical Parameter	Percentage
Sensitivity	100%
Specificity	62.2%
Positive predictive value	59.3%
Negative predictive value	100%

**Table 3 diagnostics-15-02128-t003:** Statistical analysis for *van* genes detection among the included samples.

Statistical Parameter	Percentage
Sensitivity	100%
Specificity	74.2%
Positive predictive value	22.6%
Negative predictive value	100%

**Table 4 diagnostics-15-02128-t004:** Detected resistance markers after the Allplex Entero-DR assay kit on the included samples.

Resistance Markers	Times of Detection (%) *	Culture Result (%) **
*bla* _CTX-M_	51 (29.8)	5 (9.8%)
*vanA*	22 (12.9)	10 (45.4%)
*bla*_CTX-M_ + *vanA*	22 (12.9)	10 VRE (45.4%); 1 case with VRE and CTX-M-producing strains (4.5%)
*vanB*	16 (9.3)	0
*bla*_NDM_ + *bla*_OXA-48_ + *bla*_CTX-M_	7 (4.1)	5 (71.4%)
*bla*_NDM_ + *bla*_OXA-48_ + *bla*_CTX-M_ + *vanA*	6 (3.5)	5 (83.3%)
*bla*_NDM_ + *bla*_KPC_ + *bla*_OXA-48_ + *bla*_CTX-M_ + *vanA*	5 (2.9)	0
*bla*_KPC_ + *bla*_CTX-M_ + *vanA*	5 (2.9)	1 VRE (20%); 1 CTX-M-producing strain (20%)
*bla* _KPC_	4 (2.3)	4 (100%)
*bla*_KPC_ + *bla*_CTX-M_	4 (2.3)	4 KPC-producing strains (100%)
*bla_KPC_ + vanA*	4 (2.3)	2 VRE (50%); 1 KPC-producing strain (25%); 1 case with KPC-producing and VRE strains (25%)
*bla*_CTX-M_ + *vanB*	3 (1.7)	0
*vanA + vanB*	3 (1.7)	1 VRE (33.3%)
*bla*_KPC_ + *bla*_CTX-M_ + *vanB*	2 (1.2)	1 KPC-producing strain (50%)
*bla*_NDM_ + *bla*_OXA-48_ + *bla*_CTX-M_ + *bla*_VIM_	2 (1.2)	1 CTX-M-producing strain (50%)
*bla*_NDM_ + *bla*_CTX-M_ + *vanA*	2 (1.2)	1 NDM-producing strain (50%)
*vanA + vanB* + *bla*_CTX-M_	2 (1.2)	1 CTX-M-producing strain (50%)
*bla* _VIM_	2 (1.2)	0
*bla*_KPC_ + *bla*_CTX-M_ + *vanA* + *bla*_VIM_	1 (0.6)	0
*bla*_NDM_ + *bla*_CTX-M_ + *vanB*	1 (0.6)	0
*bla*_NDM_ + *bla*_KPC_ + *bla*_CTX-M_ + *vanA*	1 (0.6)	0
*bla*_NDM_ + *bla*_KPC_ + *bla*_OXA-48_ + *bla*_CTX-M_	1 (0.6)	1 CTX-M-producing strain (50%)
*bla* _NDM_	1 (0.6)	1 NDM-producing strain (100%)
*bla*_NDM_ + *bla*_OXA-48_ + *bla*_CTX-M_ + *vanB*	1 (0.6)	1 CTX-M-producing strain (50%)
*bla*_KPC_ + *bla*_CTX-M_ + *bla*_OXA-48_	1 (0.6)	0
*bla*_OXA-48_ + *bla*_CTX-M_ + *vanA*	1 (0.6)	1 OXA-48-producing strain (100%)
*bla*_VIM_ + *vanB*	1 (0.6)	1 VIM-producing strain (100%)

* Times of detection were calculated reporting 171 positive Allplex Entero-DR assay molecular results as a common total number. ** Culture positive results were calculated reporting times of detection as a common total number. Abbreviations: VRE = vancomycin-resistant enterococci; KPC = Klebsiella pneumoniae carbapenemases; VIM = Verona integron-encoded metallo-ß-lactamase; OXA-48 = oxacillinases β-lactamases.

## Data Availability

The original contributions presented in this study are included in this article. Further inquiries can be directed to the corresponding author(s).

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
