# Peer review of "Active Microbiological Surveillance for Contrasting Multi-Drug-Resistant Pathogens: Comparison Between a Multiplex Real-Time PCR Method and Culture"

_diagnostics, 2025, doi:10.3390/diagnostics15172128_

Round 1

Reviewer 1 Report

Comments and Suggestions for Authors

The study compared the utility of commercial kits with the culture based methods for detecting ARGs and the resistant bacteria harbouring them in rectal swabs. The authors conclude that the molecular methods should be combined with conventional methods for the detection of MDR pathogens.

The title is confusing, and doesn’t clearly represent the contents of the manuscript.

Abstract needs improvement. As such, the manuscript has numerous grammatical (language) issues, which require the authors’ attention.

Some words such as “..precise genes” (L19),  double-carbapenemases’ (L29), producing strains’ (L30),  higher samples’ (L32), etc might indicate use of certain writing tools. These do not represent standard way of writing scientific manuscripts. Such punctuation marks (apostrophe) are distributed throughout the manuscript.

L30: What do you mean by “… producing strains’ diffusion”?

Section 2.2. Please provide reference for this isolation protocol.

L171: Analysis globally…. Please rephrase the sentence.

Please replace Graph 2 with Figure 2.

L171: 188 samples (62.7%) resulted positive for at least one Allplex target. What are those targets?

186: “positive for at least one carbapenemases gene”. Again what are those?

Table 1. The positive predictive values seems to be too low. How does this correlate with the manufacturer’s claims?

The culture positivity for ESBL bacteria is also too low. Would a brief pre-enrichment have increased the culture positivity? Why didn’t the authors use ESBL chromogenic agar for the isolation of ESBL-producing bacteria?

Table 4. Results in the third column do not match the results in the first column interms of genotypes identified. For example, in the second row, you have indicated CTX-M+ as 29.8%. In the third column, 9.8% is indicated. Did you isolate CTX-M+ cultures from 5 samples?

Similarly, in row number 7, blaNDM + blaOXA-48 + blaCTX-M + vanA   was 3.5% and you isolated from 5 samples. Now, were all these islolates blaNDM + blaOXA-48 + blaCTX-M + vanA ?. Or at least one each of these?

In the subsequent rows in the of the column number 3, only ESBL is indicated. Which of those ESBLs you tested by real-time PCR were actually detected in these isolates?

L296: “processed samples”?

L291-300: This paragraph requires rewriting to deliver the correct and clear message.

Comments on the Quality of English Language

The language and the presentation of the manuscript must be improved. 

Author Response

Comments and Suggestions for Authors

The study compared the utility of commercial kits with the culture-based methods for detecting ARGs and the resistant bacteria harbouring them in rectal swabs. The authors conclude that the molecular methods should be combined with conventional methods for the detection of MDR pathogens.

Comment: The title is confusing, and doesn’t clearly represent the contents of the manuscript.
Answer: The title has been slightly changed. 

Comment: Abstract needs improvement. As such, the manuscript has numerous grammatical (language) issues, which require the authors’ attention.
Answer: The abstract has been revised. Additionally, we focused our attention on the English language, modifying some sentences.

Comment: Some words such as “…precise genes” (L19), double-carbapenemases’ (L29), producing strains’ (L30), higher samples’ (L32), etc might indicate use of certain writing tools. These do not represent standard way of writing scientific manuscripts. Such punctuation marks (apostrophe) are distributed throughout the manuscript.
Answer: We used the expression “precise gene” to indicate a specific gene. On the other hand, “carbapenemases-producing strains” and “double-carbapenemases” are extremely diffused expressions in the microbiological field, indicating multi-drug-resistant strains and their corresponding features. We don’t understand the exact meaning of your comment, especially when you mention writing tools, which were not applied for none of our current or previous manuscripts.  

Comment: L30: What do you mean by “… producing strains’ diffusion”?
Answer: The complete sentence (“Our results underlined the ubiquitous blaCTX-M and van genes presence and demonstrated double-carbapenemases’ genes and metallo-β-lactamases-producing strains’ diffusion”) indicates the possibility to study resistant strains (i.e. metallo-β-lactamases-producing strains) diffusion among critical hospital settings.

Comment: Section 2.2. Please provide reference for this isolation protocol.
Answer: They have been added.

Comment: L171: Analysis globally…. Please rephrase the sentence.
Answer: The sentence has been modified.

Comment: Please replace Graph 2 with Figure 2.
Answer: It has been modified.

Comment: L171: 188 samples (62.7%) resulted positive for at least one Allplex target. What are those targets?
Answer: Table 4 illustrates the detected genes. The Allplex kit included all the different carbapenemases, CTX-M, and van genes: the details about their single or simultaneous presence among the tested samples is summarized within the table.  

Comment: 186: “positive for at least one carbapenemases gene”. Again what are those?
Answer: According to the previous comment, we included those details within the Table 4.

Comment: Table 1. The positive predictive values seems to be too low. How does this correlate with the manufacturer’s claims?
Answer: Unfortunately, this is one of the numerous studies comparing highly sensitive methodologies (molecular assays) to culture-based protocols, which are known for low sensitivity and specificity rates. The necessity to compare the Allplex kits to the gold standard method (culture) affected the statistical analysis, leading to discouraging positive predictive values. We supported these results through previously published data and included some comments within the discussion:

Specifically, the culture was often negative for the presence of the microorganism, while the molecular methods were able to detect all resistance genes. Previously published data registered similar disadvantages in comparing such different methodologies, especially about carbapenemases’ genes [11-12; 17]”.

Comment: The culture positivity for ESBL bacteria is also too low. Would a brief pre-enrichment have increased the culture positivity? Why didn’t the authors use ESBL chromogenic agar for the isolation of ESBL-producing bacteria?
Answer: The ESBL chromogenic agar is not currently included within our diagnostic routine, but we are planning to evaluate its effectiveness in the next future. As regards the pre-enrichment, we tried to enrich all the samples with Brain Heart infusion broth (data not shown), obtaining the same result. According to the absence of a further contribution to the definitive results, we decided to not include those data within the manuscript.

Comment: Table 4. Results in the third column do not match the results in the first column in terms of genotypes identified. For example, in the second row, you have indicated CTX-M+ as 29.8%. In the third column, 9.8% is indicated. Did you isolate CTX-M+ cultures from 5 samples?
Answer: The second column indicates the total number of molecular detection for the specific target, while the third column reports culture’s positive results among the positive molecular samples. Consequently, you observation is right: we isolated 5 CTX-M producing strains during the culture procedure.

Comment: Similarly, in row number 7, blaNDM + blaOXA-48 + blaCTX-M + vanA   was 3.5% and you isolated from 5 samples. Now, were all these isolates blaNDM + blaOXA-48 + blaCTX-M + vanA? Or at least one each of these?
Answer: All the 5 isolates reported all the detected targets. We found 5 blaNDM + blaOXA-48 + blaCTX-M + vanA producing strains, which is not uncommon within our geographical area. Unfortunately, Southern Italy hold a concerning primate in Europe for double-carbapenemases’ genes and glycopeptides-resistant enterococci.

Comment: In the subsequent rows in the column number 3, only ESBL is indicated. Which of those ESBLs you tested by real-time PCR were actually detected in these isolates?
Answer: That was a typo. We corrected the generic “ESBL” word with “CTX-M”.

Comment: L296: “processed samples”?
Answer: We replaced the word “processed” with the word “tested”.

Comment: L291-300: This paragraph requires rewriting to deliver the correct and clear message.
Answer: The paragraph has been revised.

Please, find all the requested corrections highlighted in yellow within the manuscript.

Reviewer 2 Report

Comments and Suggestions for Authors

The manuscript addresses rapid identification of multi-drug-resistant (MDR) pathogens, which is one of the most relevant and urgent clinical issues for infection control and antimicrobial stewardship. The comparison between Real-Time PCR and conventional culture is valuable, as it allows assessment of diagnostic speed, sensitivity, and specificity in antimicrobial surveillance. The authors should consider the following suggestions for further improvement.

  • Rephrase the first sentence in the introduction (Lines 41-43)
  • Rephrase the sentence in lines 57 to 63 in the introduction part.
  • Graph 3 should be a part of the methods or results
  • Consider applying an inferential statistical test to your data, such as chi-square or any relevant test. The study would benefit from more detailed statistical analysis comparing the two methods, including sensitivity, specificity, positive predictive value (PPV), and negative predictive value (NPV).
  • Clarification on cost-effectiveness and resource feasibility would help assess the scalability of implementing Multiplex PCR in routine surveillance.
  • The conclusion should be written under a separate heading for conclusion
Comments on the Quality of English Language

Some sentences should be paraphrased

Author Response

Comments and Suggestions for Authors

The manuscript addresses rapid identification of multi-drug-resistant (MDR) pathogens, which is one of the most relevant and urgent clinical issues for infection control and antimicrobial stewardship. The comparison between Real-Time PCR and conventional culture is valuable, as it allows assessment of diagnostic speed, sensitivity, and specificity in antimicrobial surveillance. The authors should consider the following suggestions for further improvement.

  • Comment: Rephrase the first sentence in the introduction (Lines 41-43)
    Answer: the sentence has been revised.
  • Comment: Rephrase the sentence in lines 57 to 63 in the introduction part.
    Answer: the sentence has been revised.
  • Comment: Graph 3 should be a part of the methods or results
    Answer: Thank you for the observation. We applied the requested modification.
  • Comment: Consider applying an inferential statistical test to your data, such as chi-square or any relevant test. The study would benefit from more detailed statistical analysis comparing the two methods, including sensitivity, specificity, positive predictive value (PPV), and negative predictive value (NPV).
    Answer: Thank you for this comment. We applied the Chi-square test, confirming a significance for our results’ distribution. We added some information on material and methods, along with two sentences along the results’ section.
  • Comment: Clarification on cost-effectiveness and resource feasibility would help assess the scalability of implementing Multiplex PCR in routine surveillance.
    Answer: Thank you for the observation. Actually, a cost-effectiveness analysis is a future plan for our institution. We did not produce this analysis in the current manuscript due to the non-interventional type of the study and the absence of an ethical authorization.
  • Comment: The conclusion should be written under a separate heading for conclusion
    Answer: we created a specific paragraph for conclusions.

Please, find all the requested modifications highlighted in yellow.

Round 2

Reviewer 1 Report

Comments and Suggestions for Authors

L1: Please correct as “precise identification of genes”

L41: Please rephrase- The spread of multidrug-resistant (MDR) microorganisms is one…”

L51: ..have emerged as …

L53: Glycopeptide resistance

L61: “…MDR microorganism, allowing identification of colonies and..”

L62:  Please delete “punctual”

L65: “does not allow detection of low-level expression of carbapenemases…

L68: Please delete “mechanisms’

L72: Please correct genes’ as genes

L90: Please correct units’ as unit

L91: “ on admission of patients”

L95: Please correct specimens’ as specimen

L113:  Gram-negatives’ as Gram-negative

L141: carbapenemases’ as carbapenemase

L167:” Alert microorganism detection”

L182:  samples’ number as number of samples

L182: these results’ distribution as “these results”

L186: Results’ distribution as “ distribution of results”. Please change the title accordingly, in the figure also

L197: carbapenemases’ genes as “carbapenemase genes”. Same in table 1 title, and in line 230

L244: microorganism’ growth as “growth of this microorganism”

L246: carbapenemases’ genes as “carbapenemase genes”

L274: prevent these bacteria’s spread as “prevent the spread of these bacteria”

L278: assays identify precise genes’ presence as “assays precise identify the presence of genes”

L307 & L312: carbapenemases’ genes as carbapenemase genes
L333: resistance markers’ presence as “the presence of resistance markers”

L351: arrange (propose?)

L375: multiple genes’ combinations as “multiple gene combinations”

L377: colonized patients’ tracking as “ tracking of colonized patients”

L378: pathogens’ diffusion as “pathogen spread”

L381: infections’ rate as “infection rate”
L388: higher clinical isolates’ number, as “higher number of clinical isolates

Comments on the Quality of English Language

I have indicated the corrections  to be made in my report. Further, authors are advised to get the language check done for the manuscript

Author Response

Comment: L1: Please correct as “precise identification of genes”
Answer: It has been corrected.

Comment: L41: Please rephrase- The spread of multidrug-resistant (MDR) microorganisms is one…”
Answer: It has been rephrased.

Comment: L51: have emerged as …
Answer: It has been corrected.

Comment: L53: Glycopeptide resistance
Answer: It has been corrected.

Comment: L61: “…MDR microorganism, allowing identification of colonies and..”
Answer: It has been corrected.

Comment: L62:  Please delete “punctual”
Answer: It has been deleted.

Comment: L65: “does not allow detection of low-level expression of carbapenemases…
Answer: It has been corrected.

Comment: L68: Please delete “mechanisms’
Answer: It has been deleted.

Comment: L72: Please correct genes’ as genes
Answer: It has been corrected.

Comment: L90: Please correct units’ as unit
Answer: It has been corrected.

Comment: L91: “on admission of patients”
Answer: It has been corrected.

Comment: L95: Please correct specimens’ as specimen
Answer: It has been corrected.

Comment: L113:  Gram-negatives’ as Gram-negative
Answer: It has been corrected.

Comment: L141: carbapenemases’ as carbapenemase
Answer: It has been corrected.

Comment: L167:” Alert microorganism detection”
Answer: It has been corrected.

Comment: L182:  samples’ number as number of samples
Answer: It has been corrected.

Comment: L182: these results’ distribution as “these results”
Answer: It has been corrected.

Comment: L186: Results’ distribution as “ distribution of results”. Please change the title accordingly, in the figure also
Answer: They have been corrected.

Comment: L197: carbapenemases’ genes as “carbapenemase genes”. Same in table 1 title, and in line 230
Answer: They have been corrected.

Comment: L244: microorganism’ growth as “growth of this microorganism”
Answer: It has been corrected.

Comment: L246: carbapenemases’ genes as “carbapenemase genes”
Answer: It has been corrected.

Comment: L274: prevent these bacteria’s spread as “prevent the spread of these bacteria”
Answer: It has been corrected.

Comment: L278: assays identify precise genes’ presence as “assays precise identify the presence of genes”
Answer: It has been corrected.

Comment: L307 & L312: carbapenemases’ genes as carbapenemase genes
Answer: They have been corrected.

Comment: L333: resistance markers’ presence as “the presence of resistance markers”
Answer: It has been corrected.

Comment: L351: arrange (propose?)
Answer: It has been corrected.

Comment: L375: multiple genes’ combinations as “multiple gene combinations”
Answer: It has been corrected.

Comment: L377: colonized patients’ tracking as “tracking of colonized patients”
Answer: It has been corrected.

Comment: L378: pathogens’ diffusion as “pathogen spread”
Answer: It has been corrected.

Comment: L381: infections’ rate as “infection rate”
Answer: It has been corrected.

Comment: L388: higher clinical isolates’ number, as “higher number of clinical isolates”
Answer: It has been corrected.

Please, find all the requested corrections highlighted in yellow within the manuscript. Furthermore, we wish to thank you for the observations about the English language. According to them, we already corrected some sentences with native language speakers.